# The Relationship Between Korean Adolescents’ Happiness and Depression: The Mediating Effect of Teacher Relationships and Moderated Mediation of Peer and Parental Relationships and Parental Attitudes

**DOI:** 10.3390/healthcare13070730

**Published:** 2025-03-25

**Authors:** Sookyung Jeong, Shin-Il Lim

**Affiliations:** 1Department of Nursing, College of Medicine, Wonkwang University, Iksan 54538, Republic of Korea; 2College of Nursing, Jesus University, Jeonju 54989, Republic of Korea; imsi@jesus.ac.kr

**Keywords:** happiness, adolescent, depression, parent–child relationships, school teachers, mediation analysis

## Abstract

This study explored how Korean students’ happiness impacts depression during their transition from middle to high school, emphasizing teacher relationships as a mediator and peer and parental relationships as moderators. Utilizing data from 2147 students (1150 male, 997 female) in the Korean Children and Youth Panel Survey, this study assessed happiness, depression, and relationships with teachers, peers, and parents in 2020 and 2021. Data analyses involved Pearson’s correlations, descriptive statistics, and the SPSS 28.0 macro-PROCESS model for mediation and moderated mediation. Happiness in third-grade middle school students (*M* = 3.0509, *SD* = 0.4583) was positively linked to high school teacher relationships (β = 0.1958, *p* < 0.001) and negatively linked to depression (β = −0.1732, *p* < 0.001). Teacher relationships mediated the link between happiness and depression, with an indirect effect of β = −0.0339 (*p* < 0.001). Reduced negative parental attitudes strengthened the link between happiness and teacher relationships (β = −0.1045, *p* < 0.01). Teacher–student relationships are vital for adolescent emotional health, particularly during academic stress. Policies should encourage such relationships, enhance parenting, and develop students’ social skills.

## 1. Introduction

Happiness refers to an individual’s subjective satisfaction with their life and emotional well-being [1]. Rather than being a transient emotion, it reflects a sustained self-accepting attitude in daily life. Low levels of happiness are closely associated with numerous problem behaviors among adolescents, including depression, school violence, delinquency, and gaming addiction. According to one study [2], Korean adolescents ranked last among the 22 OECD countries in subjective happiness, with levels decreasing as they progress to higher years. Notably, happiness experienced a sharp decline during students’ middle school years and continued to decline steadily into high school. This phenomenon is commonly observed not only in the Republic of Korea but also in most countries, with a more pronounced effect among adolescents in East Asian countries such as Japan and China [3]. Alongside the low levels of happiness, the incidence of depressive symptoms among Korean adolescents is gradually increasing. According to the 2022 Youth Statistics, the levels of depression among middle and high school students have consistently risen since 2020 [4]. Furthermore, the suicide rate among Korean adolescents has been increasing; in 2020, the number of adolescent suicides was 957, an increase of 81 (9.2%) from the previous year. In 2022, suicides accounted for 42.3% of the causes of death among children and adolescents, categorized by age [4].

Given that depression and happiness during adolescence can have lasting effects on an individual’s life in adulthood, recent studies have analyzed the relationship between happiness and depression during adolescence, finding that higher levels of happiness negatively influence depression [5] and that the happiness levels of depressed groups are lower than those of healthy groups [6]. Kim et al. [7] found that male students in lower grades and adolescents who were not depressed reported higher levels of subjective happiness. Furthermore, adolescent depression is negatively correlated with subjective happiness [8], and relationship conflicts influence adolescents’ subjective happiness [9].

Attachment theory, developed by John Bowlby and Mary Ainsworth, explains the formation of emotional bonds in humans [10]. This theory posits that attachment is initially formed through relationships with primary caregivers and serves as the foundation for all future interpersonal relationships. Within this framework, the teacher–student relationship during adolescence plays a crucial role in students’ emotional stability and social development. Research by Martin and Collie and Nazish et al. demonstrated that when teachers show care and respect, students experience positive emotions that contribute to school adjustment and emotional well-being [10,11]. This perspective suggests that positive teacher–student relationships significantly enhance adolescent happiness. From an attachment theory standpoint, teachers serve as emotional support figures for adolescents, and positive interactions foster the development of self-identity and social skills. This aligns with Cooke et al.’s [12] assertion that attachment relationships regulate individuals’ social behaviors and emotional responses [13]. Therefore, positive teacher–student relationships grounded in attachment theory serve as a crucial mediating factor in reducing depression and promoting happiness among students.

Adolescence is characterized by increased sensitivity and importance attached to interpersonal relationships, making this developmental period particularly pivotal [14]. During this period, friendships and social status among peers become extremely important, and adolescents seek recognition, support, and acceptance from their peers [15]. Friendships have a substantial correlation with adolescents’ happiness, comparable to that of their parental relationship [5]. Additionally, the peer environment has shown high explanatory power regarding adolescents’ happiness; positive friendships are closely linked to adolescent well-being [16]. Adolescents with strong peer attachments and amicable peer relationships reported higher life satisfaction [17], and positive correlations between happiness and support from friends have been consistently observed across all age groups and cultural contexts [18].

Parenting styles play a crucial role in helping children develop trust in their parents and others, thereby contributing to their adaptation and overall development. Positive parenting attitudes significantly affect adolescents’ adjustment, personality, and relationships during early adolescence [19]. Previous studies indicate that parental factors outweigh school or peer factors in predicting a child’s happiness [20]. Additionally, family cohesion and effective communication significantly enhance adolescent happiness [21]. Adolescents who maintain positive relationships with their parents tend to report higher levels of subjective happiness [22], while those who experience negative parenting attitudes develop unfavorable perceptions of themselves and others, which can lead to vulnerabilities such as passive relationships and social withdrawal, eventually resulting in depression. Furthermore, an association/relationship has been established between negative parenting attitudes characterized by coercion, rejection, inconsistency, and depression [23].

While previous research has explored the relationship between happiness and depression in adolescents, studies have comprehensively analyzed the mediating effect of teacher–student relationships and the moderating effects of peer and parent relationship. This gap is especially pronounced in research that considers the unique cultural context (e.g., intense academic competition, individualistic tendencies) and educational environment of Korean adolescents. Park and Noh primarily examined the mediating effects of social support from friends, family, and teachers on adolescent happiness, neglecting the moderating roles of peer and parent relationships [24,25]. This study addresses these limitations by conducting an in-depth analysis of the mediating effect of teacher–student relationships and the moderating effects of peer and parent relationships on the transition from high happiness in middle school to high depression in high school among Korean adolescents (Figure 1). By investigating how positive or negative peer and parent relationships influence the maintenance of adolescent happiness and the development of positive teacher–student relationships, this research aims to contribute to effective intervention strategies for supporting adolescent mental well-being. The research questions are as follows:What are the relationships between happiness, teacher relationships, depression, peer relationships, and parental attitudes as perceived by adolescents before and after entering high school?Do teacher relationships mediate the relationship between happiness and depression as perceived by adolescents before and after entering high school?Do peer relationships and parental attitudes exhibit moderated mediation effects on the relationship between happiness and teacher relationships as perceived by adolescents before and after entering high school?

## 2. Materials and Methods

### 2.1. Participants and Data Collection

This study utilized panel data from the Korean Children and Youth Panel Survey 2018 [26] conducted by the National Youth Policy Institute, specifically focusing on the third (middle school third year) and fourth years (high school freshman year). Data utilization adhered to the guidelines of the National Youth Policy Institute by specifying the survey name and the institute and excluding participants’ personal information. The research participants comprised 2147 individuals (1150 male and 997 female), excluding missing values from the fourth-year dataset. This longitudinal study draws on data from 2020 and 2021 and was approved by the Institutional Review Board of the National Youth Policy Institute of Korea. Data for this study, collected from tests administered by a national agency, provided reliable results and enabled the tracking of developmental processes through long-term data collection. The data were therefore deemed suitable for analyzing the longitudinal influences that are the focus of this study. The approval number for the 2020 data is 202007-HR-Goyu-016 and for the 2021 data is 202106-HR-Goyu-011.

### 2.2. Measurement Instruments

#### 2.2.1. Happiness

The happiness scale employed in this study was developed by Lee et al. [27] and has been used in the Korean Children and Youth Panel Survey [26]. It comprises four items. An example statement is “Overall, I am…” These items were rated on a 4-point Likert scale where 1 indicates very unhappy and 4 indicates very happy, with higher scores indicating greater happiness. The reliability for the third year of middle school was calculated to be 0.7963, based on the Cronbach alpha value.

#### 2.2.2. Teacher Relationships

The Student–Teacher Attachment Relationship Scale of Kim and Kim [28] was used to assess teacher relationships. This scale has been used in the Korean Children and Youth Panel Survey [26]. It consists of 14 items on accessibility and acceptability, each comprising three items, and sensitivity and reliability, each comprising four items. An example statement is “The teacher respects my opinions and allows me to speak freely”. These items were rated on a 4-point Likert scale, with higher scores indicating a stronger attachment between the teacher and the student. The reliability for the fourth year (high school freshmen) confirmed in this study was 0.9081, based on the Cronbach alpha value.

#### 2.2.3. Depression

The depression scale employed in this study, developed by Kim et al. [29], consists of 10 items and has been used in the Korean Children and Youth Panel Survey [26]. An example statement is “I feel like I want to die”. These items were rated on a 4-point Likert scale, with higher scores indicating higher levels of depression. The reliability for the fourth year (high school freshmen) verified in this study was 0.9053, based on the Cronbach alpha value.

#### 2.2.4. Peer Relationships

The peer relationship quality scale used in this study, developed by Bae et al. [30], has been utilized in the Korean Children and Youth Panel Survey [26] and comprises 13 items. Among these, eight items pertain to positive peer relationships and five items are related to negative peer relationships. An example of a positive peer relationship item is, “I spend time with friends”, while an example of a negative peer relationship item is “I frequently have conflicts of opinion with friends”. These items were rated on a 4-point Likert scale ranging from 1 (not at all) to 4 (very much so), with higher scores indicating stronger positive or negative peer relationships. The reliability of positive peer relationships for fourth-year students (high school freshmen) was 0.8537, whereas the reliability for negative peer relationships was 0.8426, as indicated by the Cronbach alpha value.

#### 2.2.5. Parental Attitudes

The Parental Style and Control Questionnaire Korean Adjustment, developed by Kim and Lee [31], was used to assess parental attitudes and has been employed in the Korean Children and Youth Panel Survey [26]. It consists of 24 items, 12 of which measure positive parenting attitudes and the other 12 negative ones. An example of a positive parenting attitude item is “My parents enjoy being with me”, and an example of a negative parenting attitude item is “My parents think I am a nuisance”. These items were rated on a 4-point Likert scale, with higher scores indicating stronger positive or negative parenting attitudes. The reliability of positive parenting attitude for the fourth year (high school freshman year) was found to be 0.9075, while the reliability for negative parenting attitude was 0.8762, based on the Cronbach alpha value.

### 2.3. Data Analysis

This study examined the mediating effect of teacher relationships on the relationship between happiness and depression among first-year high school students who graduated from middle school to verify the moderated mediation effects of peer relationships and parental attitudes on the relationship between happiness and teacher relationships. The data analysis was conducted as follows. First, frequency analysis and descriptive statistics were performed to examine the general characteristics of the participants. Second, Pearson’s correlation analysis was conducted to explore the relationships between happiness, teacher relationships, depression, peer relationships, and parental attitudes. Third, the SPSS 28.0 macro-PROCESS 4 model was used to verify the mediating effect of teacher relationships on the relationship between happiness and depression. Fourth, the moderated mediating effects of peer relationships (both positive and negative) and parental attitudes (both positive and negative) on the relationship between happiness and teacher relationships were examined. To verify the statistical significance of the mediating effect, bootstrapping was performed based on the inclusion of zero within the lower and upper bounds of the confidence interval in 10,000 samples with a 95% confidence interval as recommended by Hayes [32].

## 3. Results

### 3.1. Descriptive Statistics and Correlation Analysis

The results of the descriptive statistics and correlation analysis for the main variables of the study—happiness, teacher relationships, depression, peer relationships, and parental attitudes—are presented in Table 1. The mean scores for happiness, teacher relationships, depression, positive peer relationships, negative peer relationships, positive parental attitudes, and negative parental attitudes were 3.0509 (*SD* = 0.4583), 2.7418 (*SD* = 0.4534), 1.7846 (*SD* = 0.5550), 3.0682 (*SD* = 0.4726), 1.7918 (*SD* = 0.8745), 3.1171 (*SD* = 0.453), and 1.9634 (*SD* = 0.490), respectively. Skewness and kurtosis were examined to assess the normality of each variable. Skewness ranged from −0.2746 to 0.5207 and kurtosis from −0.1982 to 1.1595, indicating that the assumption of a normal distribution was met (skewness < 2, kurtosis < 7) [33,34]. The Variance Inflation Factor (VIF) for confirming multicollinearity ranged from 1.2941 to 1.6675, with all variables falling below 3, indicating that multicollinearity is not a significant issue [35].

The results of the correlation analysis for each variable were as follows. First, happiness in third-year middle school had a significant positive relationship with teacher relationships in the high school freshman year, positive peer relationships, and positive parental attitudes. It had a significant negative relationship with depression in the high school freshman year, negative peer relationships, and negative parental attitudes. Second, high school freshman year teacher–student relationships had a negative relationship with depression in the high school freshman year, negative peer relationships, and negative parental attitudes, and a positive relationship with positive peer relationships and positive parental attitudes. Third, depression in the high school freshman year had a negative relationship with positive peer relationships and positive parental attitudes, and a positive relationship with negative peer relationships and negative parental attitudes. Fourth, among peer relationships, positive peer relationships in the high school freshman year had a negative relationship with negative peer relationships and negative parental attitudes, and a positive relationship with positive parental attitudes. Additionally, negative peer relationships in the high school freshman year had a negative relationship with positive parental attitudes and a positive relationship with negative parental attitudes. Finally, positive parental attitudes in the high school freshman year showed a significant negative relationship with negative parental attitudes.

### 3.2. Mediating Effect of Teacher Relationships on the Relationship Between Happiness and Depression

The results of the analysis of the mediating effect of high school freshman year teacher relationships on the relationship between middle school third year happiness and depression in high school freshman year are presented in Table 2. To clarify causality, the depression levels of the middle school third-year students were controlled in the analysis. Consequently, middle school third year happiness influenced teacher relationships in the high school freshman year (β = 0.1470, *p* < 0.001), and teacher relationships in the high school freshman year had a significant effect on depression in the high school freshman year (β = −0.1394, *p* < 0.001), indicating that teacher relationships in the high school freshman year mediated the relationship between middle school third year happiness and depression in the high school freshman year. Furthermore, as exhibited in Table 3, the total effect of middle school third year happiness on depression in the high school freshman year was β = −0.1054 (*p* < 0.001). However, when the mediating variable, teacher relationships, was included, the direct effect of middle school third year happiness on depression in the high school freshman year decreased to β = −0.0849 (*p* < 0.001), demonstrating mediation by teacher relationships. Statistically, the total indirect effect was β = −0.0205 (*p* < 0.001), with neither the upper nor lower bounds including 0, thus indicating significance.

### 3.3. Moderating Effect of Peer Relationships (Positive/Negative) and Parental Attitudes (Positive/Negative) on Happiness and Teacher Relationships

The results of the moderating effect of peer relationships in the high school freshman year (positive/negative) and parental attitudes (positive/negative) on middle school third year happiness and teacher relationships in the high school freshman year are presented in Table 4. In the relationship between middle school third year happiness and teacher relationships in the high school freshman year, neither positive nor negative peer relationships in the high school freshman year, nor positive parental attitudes in the high school freshman year showed a significant moderating effect. Parents likely provide basic emotional stability and psychological protection until adolescence, thereby facilitating emotional regulation regarding happiness and teacher relationships. However, friendships may exert a greater influence on social adaptation, sense of belonging, and the development of interpersonal skills within peer groups than on emotional stability, consistent with prior research findings [36,37]. These results suggest that adolescents’ focus varies with their developmental stage and that key psychological developmental elements differ accordingly. Only negative parental attitudes in the high school freshman year demonstrated a significant moderating effect (β = −0.1045, *p* < 0.01). This suggests that the impact of perceived happiness on teacher relationships varies depending on the level of negative parental attitudes. Additionally, with the addition of the interaction term, the change in R^2^ was 0.0030 (*p* < 0.01), which statistically confirmed the moderating effect of negative parental attitudes on the relationship between middle school third year happiness and teacher relationships in the high school freshman year.

These findings suggest that the strength of the association between middle school happiness and high school teacher–student relationships is moderated by the quality of parent–child relationships. Specifically, adolescents reporting negative parent–child relationships exhibited a stronger positive correlation between middle school happiness and high school teacher–student relationships (β = [insert beta value], *p* < [insert *p*-value]), compared to those reporting positive parent–child relationships. This pattern is consistent with the risk buffering model [38] and attachment theory [39], suggesting that adolescents lacking sufficient positive parental support seek compensatory support from positive teacher–student relationships.

Since the moderating effect of negative parental attitudes was significant, its moderating effect by level was analyzed. The results are summarized in Table 5. When negative parental attitudes were classified as low, medium, or high, based on the mean, the effect of adolescents’ happiness on their teacher relationships decreased as negative parental attitudes increased. As shown in Figure 2, the slope increased, indicating an increased influence of middle school third year happiness on teacher relationships, as negative parental attitudes decreased.

## 4. Discussion

This study investigated the mediating role of teacher–student relationships and the moderating roles of peer and parental relationships in the transition from high levels of happiness in middle school to high levels of depression in high school. Specifically, considering the high academic stress and competitive educational environment experienced by Korean adolescents, this study examined both the beneficial effects of positive teacher–student relationships on emotional well-being and the detrimental effects of negative parental attitudes. The results indicated that high school teacher–student relationships mediated the association between middle school happiness and high school depression. Furthermore, negative parental attitudes moderate the relationship between middle school happiness and high school teacher–student relationships; specifically, the influence of happiness on teacher–student relationships diminishes as negative parental attitudes increases.

Republic of Korean society places a strong emphasis on university graduation as a prerequisite for social mobility. Consequently, high school students experience significantly greater academic pressure and stress than their middle school counterparts, contributing to higher rates of depression and its severity. This phenomenon is not unique to Republic of Korea; adolescents in other highly competitive East Asian education systems, such as those in China, Japan, and India, also exhibit increasing depression rates as they advance in school, largely due to a societal and cultural emphasis on education [40]. Compared to their Western counterparts in the USA, Germany, and Australia, East Asian adolescents experienced greater academic pressure and stress, stemming from Confucian values that promote high educational expectations and view educational success as a crucial pathway to social advancement [41]. In this context, the quality of the teacher–student relationship during adolescence significantly influences academic achievement and emotional well-being, as demonstrated by numerous prior studies [42]. Positive teacher–student relationships served as a protective factor against depression and anxiety. Given the substantial time that adolescents spend in school, teachers assume a vital role as emotional support figures, comparable to parents. Therefore, careful observation of students’ emotional responses and timely interventions proved essential for preventing adolescent depression. The teacher–student relationship functioned as both a predictive and protective factor in adolescents’ emotional challenges [43].

Numerous studies have demonstrated the critical role of the relationship between adolescents and their teachers. In particular, a positive relationship with teachers not only influences the achievement of high academic success but also emotional support [12,43,44]. Positive relationships and emotional support from teachers reduce depression and anxiety among adolescents. As students spend more time at school with increasing years, teachers become key figures in their lives, comparable to parents. Therefore, the relationship with teachers is a predictive and protective factor for adolescents’ emotional challenges [45]. Carefully monitoring and focusing on students’ emotional responses are crucial to prevent adolescent depression.

This study found that negative parenting attitudes act as a moderating variable in the relationship between the happiness of third year middle school students and their relationships with teachers in their first year of high school, such that the more negative the parenting attitude, the less impact adolescent happiness has on teacher relationships. Consequently, it was observed that the happiness perceived by adolescents positively influenced their relationships with teachers when there were fewer negative parenting attitudes. According to Choi [46], students who experience warmth and support for autonomy from their parents form stable relationships with their teachers, which also affect their academic achievement. However, this study found that only negative parenting attitudes exerted a significant moderating effect on adolescents’ happiness and their relationships with teachers, while positive parenting attitudes did not. This aligns with the findings of Porumbu and Necşoi [47], which suggested that negative parenting attitudes diminish emotional intimacy with teachers. Adolescents’ happiness decreases due to negative parenting attitudes, leading to a lack of emotional bond with teachers and ultimately adolescent depression. As a result, it is evident that parents’ parenting attitudes during the first year of high school have a substantial impact on adolescents’ emotions. Numerous studies have suggested that parental education programs are effective in improving parenting attitudes, increasing democratic and affectionate attitudes of parents and decreasing their indifferent attitudes [48]; these improvements in parent–child communication and changes in parental attitudes then ultimately prevent potential problems in children [49]. Therefore, given the significant impact of parenting attitudes on the emotional aspects of adolescents during the first year of high school, it is necessary for schools to provide various programs, such as parental education seminars, parent coaching programs, and family education programs to enhance parent–child relationships.

This study found that peer relationships in the third year of middle school did not have a moderating effect on teacher relationships in the first year of high school. This may be because friendships in the third year of middle school do not often continue into the first year of high school, resulting in reduced influence. In the Republic of Korea, many adolescents change schools when transitioning from middle to high school, making it challenging to maintain their earlier close friendships, which might have led to these findings. However, peer relationships during the first year of high school were correlated with teacher relationships, such that better peer relationships were associated with better teacher relationships, whereas negative peer relationships were associated with poorer teacher relationships. Peer relationships become more important during adolescence than during childhood, and adolescents tend to rely heavily on them psychologically [50]. It is known that positive peer relationships enhance one’s sense of self-worth and belonging and positively affect relationships with teachers [51]. Additionally, positive interactions with peers can positively influence interactions with teachers, fostering better teacher–student relationships [52]. However, during this period, adolescents are easily influenced by the values, attitudes, and behaviors of their peers, receiving both negative and positive effects [53]. Therefore, efforts from families and schools are needed to provide programs or activities that help adolescents form healthy and positive peer relationships during this critical period.

This study relied on self-reported data, which may have been subject to response bias due to adolescents’ subjective perceptions. Furthermore, the cross-sectional design limited the ability to definitively establish causal relationships. Although the measurement instruments demonstrated acceptable reliability and validity, future research could enhance these aspects by employing more comprehensive scales or alternative measurement approaches. The sample, being limited to Korean adolescents, may restrict the generalizability of the findings to other cultural contexts. Finally, uncontrolled variables might have influenced the results. Future research should address these limitations by employing longitudinal designs to track changes over time, using more robust measurement tools, and expanding the sample to include diverse cultural groups. Further investigations could also explore additional mediating and moderating variables, such as school climate, social support networks, and coping mechanisms. Qualitative research methods could complement quantitative findings to provide a richer understanding of the complex interplay between factors influencing adolescent well-being.

## 5. Conclusions

This study highlights the critical role of teacher relationships in mitigating the impact of happiness-related decline on adolescent depression, demonstrating the potential for teachers to provide emotional support during the stressful transition from middle school to high school. The results highlight that strong teacher–student relationships serve as a buffer against the increase in depression typically observed during this period, particularly from cases of negative parental attitudes. The findings suggest that educational policies should focus on fostering positive teacher–student interactions, while schools should offer programs to improve parenting practices and develop students’ social skills. Addressing these issues could significantly enhance adolescents’ happiness and resilience to depression.

## Figures and Tables

**Figure 1 healthcare-13-00730-f001:**
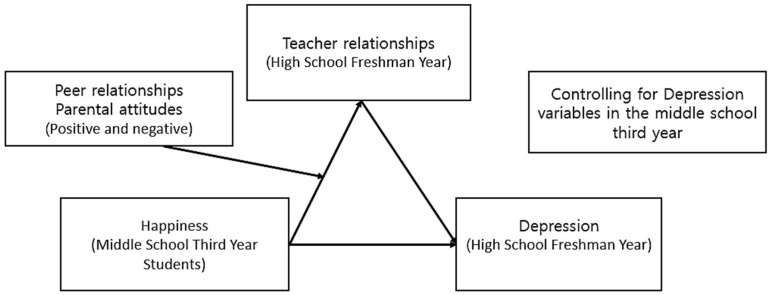
Research model.

**Figure 2 healthcare-13-00730-f002:**
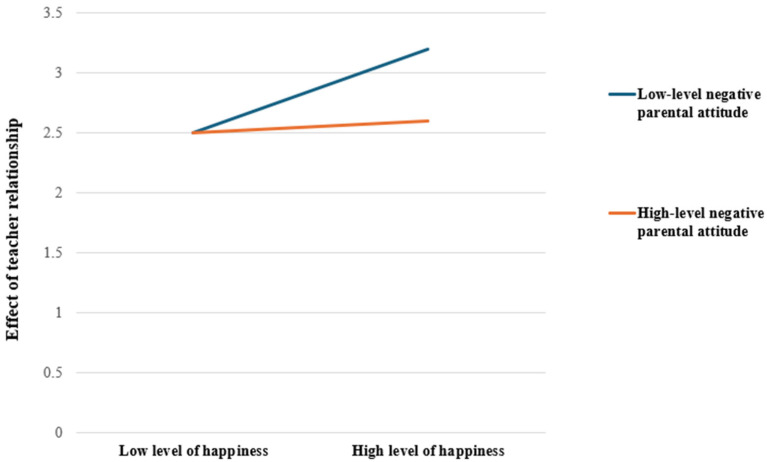
Moderating effect of negative parental attitude on the relationship between happiness and teacher relationship.

**Table 1 healthcare-13-00730-t001:** Correlations and descriptive statistics of Korean adolescents’ happiness, teacher relationship, depression, peer relationship, and parental attitudes.

	1.	2.	3.	4.	5.	6.	7.
1. Happiness of Middle School Third-Year Students	1						
2. Teacher Relationships in High School Freshman year	0.1979 ***	1					
3. Depression in High School Freshman year	−0.3166 ***	−0.1986 ***	1				
4. Peer Relationship in High School Freshman year (Positive)	0.1738 ***	0.3446 ***	−0.2594 ***	1			
5. Peer Relationship in High School Freshman year (Negative)	−0.1418 ***	−0.1538 ***	0.4227 ***	−0.2778 ***	1		
6. Parental Attitude in High School Freshman year (Positive)	0.2826 ***	0.4592 ***	−0.3345 ***	0.3877 ***	−0.2657 ***	1	
7. Parental Attitude in High School Freshman year (Negative)	−0.2302 ***	−0.1885 ***	0.4125 ***	−0.1797 ***	0.4502 ***	−0.4462 ***	1
Mean (*M*)	3.0509	2.7418	1.7846	3.0682	1.7918	3.1171	1.9634
Standard Deviation (*SD*)	0.4583	0.4534	0.5550	0.4726	0.8745	0.453	0.490
Skewness	−0.1214	−0.2689	0.4404	−0.1779	0.5207	−0.2746	0.3510
Kurtosis	1.0255	1.1595	−0.1982	0.6423	0.2201	0.6755	0.2432
Variance Inflation Factor (VIF)	1.1683	1.3306	1.4337	1.2941	1.4180	1.6675	1.5466

*** *p* < 0.001.

**Table 2 healthcare-13-00730-t002:** Mediating effect of year 1 teacher relationship on the relationship between year 3 happiness and year 1 depression.

Dependent Variable	Independent Variable	β	*SE*	*t*	LLCI	ULCI
Teacher Relationship in High School Freshman year	Constant	2.1453	0.0987	24.4646 ***	2.2217	2.6089
	Happiness of Middle School Third-Year Student	0.1470	0.0249	5.9149 ***	0.0983	0.1957
Depression in High School Freshman year	Constant	1.7481	0.1220	14.3227 ***	1.5087	1.9874
	Happiness of Middle School Third-Year Student	−0.0849	0.0274	−3.1019 ***	−0.1387	−0.0312
	Teacher Relationship in High School Freshman year	−0.1394	0.0236	−5.9050 ***	−0.1857	−0.0931

*** *p* < 0.001. LLCI = Lower limit of confidence interval for bootstrapped indirect effects. ULCI = Upper limit of confidence interval for bootstrapped indirect effects.

**Table 3 healthcare-13-00730-t003:** Indirect effect of teacher relationships on the relationship between happiness and depression.

Path	β	Standard Error (*SE*)	95% Confidence Interval (CI)	
			LLCI	ULCI
Total Effect	−0.1054	0.0274	−0.1591	−0.0517
Direct Effect	−0.0849	0.0274	−0.1387	−0.0312
Total Indirect Effect	−0.0205	0.0053	−0.0322	−0.0110

LLCI = Lower limit of confidence interval for bootstrapped indirect effects. ULCI = Upper limit of confidence interval for bootstrapped indirect effects.

**Table 4 healthcare-13-00730-t004:** Indirect effects of peer relationships (positive and negative) and parental attitude (positive and negative) on happiness and teacher relationships.

Moderating Variable	Variable	β	*SE*	*t*	*p*	95% Confidence Interval (CI)	
						LLCI	ULCI
Peer Relationship in High School Freshman Year (Positive)	Constant	2.7407	0.0092	297.6522	0.0000	2.7226	2.7587
	Happiness of Middle School Third-Year Students	0.1404	0.0201	6.9714	0.0000	0.1009	0.1800
	Peer Relationship in High School Freshman Year (Positive)	0.3064	0.0195	15.6798	0.0000	0.2681	0.3447
	Interaction	0.0296	0.0393	0.7519	0.4522	−0.0475	0.1067
	Increase in *R*^2^ due to Interaction	*R* ^2^		*F*		*p*	
		0.0002		0.5654		0.4522	
Peer Relationship in High School Freshman Year (Negative)	Constant	2.7393	0.0096	285.0916	0.0000	2.7205	2.7581
	Happiness of Middle School Third-Year Students	0.1754	0.0210	8.3493	0.0000	0.1342	0.2166
	Peer Relationship in High School Freshman Year (Negative)	−0.0996	0.0167	−5.9475	0.0000	−0.1325	−0.0668
	Interaction	−0.0660	0.0367	−1.7951	0.0728	−0.1380	0.0061
	Increase in *R*^2^ due to Interaction	*R* ^2^		*F*		*p*	
		0.0014		3.2224		0.0728	
Parental Attitude in High School Freshman Year (Positive)	Constant	2.7432	0.0090	305.9132	0.0000	2.7256	2.7608
	Happiness of Middle School Third-Year Students	0.0749	0.0199	3.7632	0.0002	0.0359	0.1139
	Parental Attitude in High School Freshman Year (Positive)	0.4387	0.0199	21.9901	0.0000	0.3995	0.4778
	Interaction	−0.0239	0.0390	−0.6121	0.5405	−0.1004	0.0527
	Increase in *R*^2^ due to Interaction	*R* ^2^		*F*		*p*	
		0.0001		0.3747		0.5405	
Parental Attitude in High School Freshman Year (Negative)	Constant	2.7364	0.0097	282.0791	0.0000	2.7173	2.7554
	Happiness of Middle School Third-Year Students	0.1567	0.0213	7.3443	0.0000	0.1148	0.1985
	Parental Attitude in High School Freshman Year (Negative)	−0.1389	0.0198	−6.9998	0.0000	−0.1778	−0.1000
	Interaction	−0.1045	0.0402	−2.5993	0.0094	−0.1833	−0.0257
	Increase in *R*^2^ due to Interaction	*R* ^2^		*F*		*p*	
		0.0030		6.7564		0.0094 **	

** *p* < 0.01. LLCI = Lower limit of confidence interval for bootstrapped indirect effects. ULCI = Upper limit of confidence interval for bootstrapped indirect effects.

**Table 5 healthcare-13-00730-t005:** Conditional indirect effects based on parental attitude (negative).

Baseline		Effect	Standard Error (*SE*)	95% Confidence Interval (CI)	
				LLCI	ULCI
Parental Attitude (Negative)	−1*SD* (−0.4910)	−0.0360	0.0078	−0.0522	−0.0218
	*M* (0)	−0.0271	0.0058	−0.0394	−0.0164
	+1*SD* (0.4910)	−0.0182	0.0063	−0.0313	−0.0068

LLCI = Lower limit of confidence interval for bootstrapped indirect effects. ULCI = Upper limit of confidence interval for bootstrapped indirect effects.

## Data Availability

The data used in this study are from the 2018 Child and Adolescent Panel Survey conducted by the National Youth Policy Institute of Korea. One can access the data by visiting the following link and applying for data usage with the responsible person. After obtaining permission, you will be able to download the requested data: https://www.nypi.re.kr/archive/mps/program/examinDataCode/dataDwloadAgreeView?menuId=MENU00226 (accessed on 10 March 2024).

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
