# Peer review of "The Relationship Between Korean Adolescents’ Happiness and Depression: The Mediating Effect of Teacher Relationships and Moderated Mediation of Peer and Parental Relationships and Parental Attitudes"

_healthcare, 2025, doi:10.3390/healthcare13070730_

Round 1

Reviewer 1 Report

Comments and Suggestions for Authors

1.This study intends to explore the issues related to adolescent's psychological well-being, which, without a doubt, is indeed very important.  However, the causal mechanism among the employed variables needs to be clarified furthermore based on clearly theoretical perspectives.

2.For example, the causal mechanisms among adolescent's happiness, depression and teacher relationship are not as coherent as the arguments by this study. The authors have to argue their causal arguments based on some insightful theories but not on their own subjective thinkings. Apparently, this study did not present their theoretical grounds appropriately.

3.This study did not take advantage of their panel designed data since it just put variables from different waves into the tested model and did not controlling for the condition of the same variable of previous wave(for example, in the tested model, the measurement of adolescent depression of third years in middle school should be included into the tested model as a control variable). To estimate the causal effects more accurately, this is a must to do procedure that controlling for previous wave's measure of dependent variable.

4.The analytic results are not sufficient to support the hypotheses of this study. For example, as the authors pointed out in the conclusion that " only negative parenting attitudes exerted a significant moderating effect on adolescents’ happiness and their relationships with teachers, while positive parenting attitudes did not," this situation is not consistent with their "theoretical expectations" and this inconsistent has to be interpreted in a more theory-based manner but not just mentioned that these analytic findings "align with" other study as a solid evidence.

5.It is suggested that the authors provide an explicit figure of their theoretical framework to lay out the tested model with the hypotheses all presented at the appropriated places.

6.One more suggestion that the authors might want to consider applying SEM to test their theoretical models since SEM seems to be more straightforward.

Author Response

Thank you very much for your valuable comments. We have carefully considered your suggestions and incorporated the necessary changes into the manuscript. Plese find the revised document attached for your review. We appreciate your time and efforts in improving our manuscript. 

Reviewer 2 Report

Comments and Suggestions for Authors

Comments on the Quality of English Language

 The English could be improved to more clearly express the research.

Author Response

Thank you very much for your valuable comments. We have carefully considered your suggestions and incorporated the necessary changes into the manuscript. Please find the revised document attached for your review. We appreciated your time and efforts in improving our manuscript. 

Reviewer 3 Report

Comments and Suggestions for Authors
  • Dear Authors,

    Your research is well-written, high-quality, and has significant potential practical implications for Korean adolescents. Following are my proposals for revisions:

    • In the introduction section, are required more updated references.
    • Lines 87-91: The authors have to justify the necessity of the present research in terms of existing literature.
    • Line 97. Is this the third research question? Then, put the number 3 in front of it.
    • 2.2 Measurement-instruments. The authors have to explain why they have chosen the specific research tools, among other possibly standardized ones.
    • The authors could start the discussion with a paragraph on the aims of the research. This could enhance the flow of the writing.
    • Lines 350-355 should be moved to the end of the discussion section. The limitations and proposals for future research should also be expanded.
    • There are no practical implications in the discussion section, and that part is of crucial importance.

Author Response

Thank you very much for your valuable comments. We have carefully considered your suggestions and incorporated the necessary changes into the manuscript. Please fine the revised document attached for your review. We appreciate your time and efforts in improving our manuscript. 

Round 2

Reviewer 1 Report

Comments and Suggestions for Authors

1.This revised paper has made direct and clear responses to the reviewer's questions and has clarified its theoretical arguments in a more systematic manner. 

2.The authors improved their data analysis processes a lot and made it easier to follow in a more adequate way. 

Reviewer 3 Report

Comments and Suggestions for Authors

The authors have addressed all my comments. The manuscript is significantly improved.